# Neonatal Bisphenol A Exposure Affects the IgM Humoral Immune Response to 4T1 Breast Carcinoma Cells in Mice

**DOI:** 10.3390/ijerph16101784

**Published:** 2019-05-20

**Authors:** Ricardo Hernández Avila, Margarita I. Palacios-Arreola, Karen E. Nava-Castro, Jorge Morales-Montor, Pedro Ostoa-Saloma

**Affiliations:** 1Departamento de Inmunología, Instituto de Investigaciones Biomédicas, Universidad Nacional Autónoma de Mexico, AP 70228, Ciudad de Mexico CP 04510, Mexico; riavila@biomedicas.unam.mx (R.H.A.); jmontor66@hotmail.com (J.M.-M.); 2Laboratorio de Genotoxicología y Mutagénesis Ambientales, Departamento de Ciencias Ambientales, Centro de Ciencias de la Atmósfera, Universidad Nacional Autónoma de Mexico, Ciudad de Mexico CP 04510, Mexico; mi_palacios_arreola@hotmail.com (M.I.P.-A.); karlenc@atmosfera.unam.mx ( K.E.N.-C.)

**Keywords:** BPA, IgM, 4T1 cells

## Abstract

Bisphenol A (BPA) is an endocrine disruptor of estrogenic nature. During the early stages of development, any exposure to BPA can have long-term effects. In this work, we study the potential alterations to the humoral antitumor immune (IgM) response in adult life after a single neonatal exposure to BPA. Female syngeneic BALB/c mice were exposed to a single dose of BPA of 250 μg/kg. Once sexual maturity was reached, a breast tumor was induced. After 25 days, the serum was obtained, and the populations of B cells in the spleen and lymph nodes were analyzed by flow cytometry. The reactivity of IgM was evaluated by 2D immunoblots. No significant changes were found in the B cell populations in the peripheral lymph nodes and the spleen. The level of ERα expression was not significantly different. However, the IgM reactivity was affected. In individuals treated with BPA, a decrease in the number of IgMs that recognize tumor antigens was observed. The possibility that these antibodies are the high affinity products of the adaptive response is discussed. The recognition of IgG was also evaluated but a null recognition was found in the controls as in the individuals treated with the 4T1 cells.

## 1. Introduction

The study of the humoral immune response to tumor antigens has the clinical value that antibodies can be used as tools for immunodiagnosis. However, there is currently no reliable immunodiagnostic test that can be used to determine the presence of a tumor in its initial stages. This may be explained by the alterations of the antitumor response due to environmental factors, making it difficult to identify reliable biomarkers at this stage of tumor development. These environmental factors include some pollutants, which have been described as endocrine disruptering chemicals (EDCs) [1,2]. EDCs may have estrogenic, antiestrogenic or antiandrogenic activity. These compounds are highly lipophilic and are stored for prolonged periods in adipose tissue [3]. In fact, it was observed that maternal exposure to EDCs during pregnancy and lactation allows the passage of these products through the placenta and breast milk [4]. Bisphenol A (BPA) is an organic compound used massively in the production of epoxy resins that are present in the coating of food cans and polycarbonates, which can be consumed by humans. BPA has been classified as an endocrine disruptor of estrogenic nature since it has an affinity for the estrogen receptors ERα and ERβ, whose effects vary according to the dose, tissue and stage of development of the individual [5,6]. The main source of exposure to BPA in humans is thought to occur by oral ingestion. However, transdermal absorption [7] and inhalation of contaminated dust in the air [8] are likely secondary routes.

Recently, it has been reported that the exposure of female syngenic BALB/c mice to a single dose of BPA (250 μg/kg) during the neonatal state contributes to the development and progression of breast cancer by modulating the antitumor immune response [9]. Taking into account the network of neuro-immuno-endocrine interactions, it is possible that BPA could affect the humoral immune system particularly if exposure occurs during the critical periods of development. There are studies of BPA and immune function. However, most of these studies are conducted in vitro [10]. The few in vivo studies that exist do not take into account that the immune response must be studied under antigenic challenge conditions.

We have previously characterized the humoral response of natural antibodies (IgM) in a murine cancer model using 4T1 cells [11]. As the BPA is an important EDC and its impact on immune regulation has been demonstrated, we evaluated the variation of the humoral immune response, which is caused by BPA in the B cell populations and measured this by flow cytometry. We measured the B cell function by the reactivity of IgM antibodies to tumor cell 4T1 breast cancer antigens in mice that received a single neonatal exposure to bisphenol A.

Our study establishes that there are changes in the patterns of recognition by IgM towards tumor antigens. This variation means that the individuals treated with BPA stop recognizing antigens that are normally recognized by all individuals. Reference [11] is a work that our group published, in which we describe the common antigens that are recognized by mice throughout the time that the tumor develops among other things. We found that as the tumor develops, more antigens that are common to all individuals are detected. We interpret this effect as the appearance of IgM, which is the product of the adaptive immune response of the mouse and differentiate it from the innate immune response that occurs at time zero before the application of the tumor. We believe that BPA responds to the adaptive response through the proven effects of BPA on cytokines, such as IFN-gamma [9]. Furthermore, we demonstrate the important role that IFN plays in the differentiation of B cells.

## 2. Materials and Methods

### 2.1. Ethics Statement

The procedure for animal handling is described in [9]. As an ethical statement, the technical specifications for the production, care and use of laboratory animals can be consulted in: “NORMA Oficial Mexicana NOM-062-ZOO-1999, Especificaciones técnicas para la producción, cuidado y uso de los animales de laboratorio” [12].

### 2.2. Neonatal BPA Exposure

The procedure is described in [9]. Although the main exposure route is commonly oral, subcutaneous injection was selected instead as no differences between oral and subcutaneous routes are observed in neonate mice in this case.

### 2.3. Mammary Tumour Induction

The procedure is described in [9]. For the tumor induction, we followed the method published by Pulaski and Ostrand-Rosenberg [13]. All mice that received 4T1 cells developed a tumor.

### 2.4. Cell Culture and Flow Cytometry

The procedure is described in [9]. After the cardiac puncture, the spleen and the inguinal peripheral lymph nodes were mechanically disaggregated using a 50 μm nylon mesh and washed with PBS. These organs were placed in the wells of culture plate with 1 mL of PBS and kept on ice. The nodules were kept cold in PBS until processing for flow cytometry. Spleen erythrocytes were lysed by incubation for 10 min in erythrocyte lysis buffer (0.15 M NH_4_Cl, 1 M KHCO_3_, 0.1 mM Na_2_EDTA, pH 7.3) for 10 min and washed with PBS.

### 2.5. Two-Dimensional Immunoblot and Image Processing

The 2D immunoblots and image analysis were performed as previously described [11]. An average 2D image of the duplicates of each mouse was obtained. A master image was obtained from the superposition of all the average images of all the mice under different experimental conditions. The master image was used to identify the spots that were common to all mice.

### 2.6. Statistical Analysis

The experimental design considers two independent variables: neonatal exposure (Intact, BPA vehicle) and induction of mammary tumors (Control or 4T1). Data from 2–3 independent experiments were analyzed using the Prism 6^®^ software (GraphPad Software Inc.) A one-way ANOVA (alpha = 0.05) was performed. The differences were considered to be significant if *p* < 0.05. The estrogen receptor expression data consider the two independent variables and therefore, a two-way ANOVA (alpha = 0.05) was performed, followed by a and Bonferroni multiple comparison between all groups.

## 3. Results

Using flow cytometry, a subpopulation of lymphocytes was selected from the size and complexity characteristics associated with the parameters FSC-A and SSC-A, respectively. The area corresponding to these cells was selected and the identification of the B lymphocytes solely used the location of the CD19+ population within the region corresponding to the lymphocytes, as shown in Figure 1.

The percentage of the total B lymphocyte subpopulation was analyzed by cytometric staining of the peripheral lymph nodes (PLN) and the spleen of the experimental subjects. Tumor development causes an increase in the percentage of B lymphocytes in the lymph nodes but with respect to the effect of neonatal endocrine disruption, no significant differences are observed. In the spleen, the proportion of B lymphocytes decreased in tumor-bearing animals. However, no significant differences were observed due to the effect of neonatal BPA in healthy groups or those with tumor development (Figure 2).

To determine the effects of BPA on the humoral immune system under competent immune conditions, we tested the hypothesis that the exposure of mice to BPA could affect the production of specific antibodies to tumor antigens. Figure 3 shows the reactivity of serum IgM antibodies from mice exposed to BPA, which are located on a 4T1 cell lysate separated bidimensionally. There are differences in the reactivity between the control group, the vehicle and the individuals treated with BPA. The immunoblots represent the master image obtained from average images of mice in each experimental situation. There were N = 6 for the control group, N = 4 for the vehicle group and N = 7 for the BPA group. The red boxes indicate the spots that are common to all the mice in that group. In individuals treated with BPA, a decrease in the reactivity of a population of IgM is observed. In the discussion, we will try to show that this group of IgMs could belong to a population of cells expressing high affinity IgM that are part of the adaptive humoral immune response.

The cells of the immune system are subject to modulation by sex steroids, which is mediated by receptors present in these cells. Given the estrogenic nature of the endocrine disruption caused by BPA, the expression of the ERα receptor in B lymphocytes was of interest. In the draining PLNs, we found no changes in the evaluation of ERα expression in terms of the percentage of ERα^+^ cells and the level of expression according to the median intensity of fluorescence between the experimental groups. On the other hand, in the spleen, we observed that the B lymphocytes of the animals that developed tumors had a lower expression of ERα compared to healthy individuals. This phenomenon was more evident in the groups treated with vehicle and BPA (Figure 4).

## 4. Discussion

In this work, we analyze the effect of the neonatal administration of BPA on the immune response of the B cell population. We did not find significant changes in the percentage of the B cell population when the peripheral lymph nodes and the spleen were analyzed in mice treated with BPA. This may be explained by previous studies, which demonstrated that the inoculation of 4T1 cells tends to immunosuppress the mice that receive these cells [14,15]. We observed this phenomenon when the B cells in the spleen are analyzed as shown in Figure 2. It is likely that the non-observance of some effect of BPA on the average population of B cells is due to the fact that this immunosuppression is greater than the measured effect, which subsequently masks it. It is possible that the use of another murine model of breast cancer will be required to accurately assess the effect of BPA on the percentage of the B cell population.

We did not found changes in B lymphocytes and ERα levels because it may be more appropriate to search for plasmatic cells. B cells did not change because their production may also not be affected by BPA, with its differentiation to plasmatic cells potentially being the key. We are currently working to elucidate this effect of BPA. As for the ERα, the binding of BPA probably does not affect its expression and thus, there will be no changes despite the mechanism of action occurring through that pathway. To block ER using tamoxifen or ICI-250 would help to dilucidate if the BPA mechanism of action only occurs by binding to ERα. Otherwise, it is possible that other steroid hormone receptors are involved in the BPA effect, such as PR or even ERα.

However, even when the percentages of B cells were not altered by BPA treatment, we looked for alterations in the immunoglobulin reactivity in treated animals. The responses of B-2 cells are optimized in the germinal centers (GC), which are follicular structures within the secondary and tertiary lymphoid organs [16]. The main functions of the GC reaction are the diversification of antibodies through the recombination of class change and the generation of B cells matured by affinity through the process of somatic hypermutation [17]. In addition, GCs facilitate the generation of memory B cells and produce large quantities of short-lived and long-lived plasma cells that provide high titers of antibodies in serum [18].

There are two types of IgM found under normal conditions in the circulation in mice. The first is natural IgM, which is secreted mainly by B-1a CD5^+^ cells in the apparent absence of antigenic stimulation. This natural IgM constitutes a major part of the circulating IgM. The other type of IgM is adaptive IgM, which is induced only after stimulation with antigens and is mainly produced by conventional B (B-2) cells. The recognition of IgM patterns after the inoculation of 4T1 cells is caused by both the acquired immune response and the innate immune response. One way to differentiate between the two responses involves examining the affinity of the antibodies produced. Natural IgMs are low affinity antibodies while IgMs of the adaptive response are high affinity antibodies [19,20]. The IgM response that we reported here from B cells against 4T1 cells suggest a dominant role of IFN-gamma in these processes because the humans and mice with deficiencies in the IFN-gamma/IL-12 axis have a pronounced susceptibility to infection [21,22]. In fact, it has been reported that the GC reaction that generates B cells of memory IgM^+^ against *S. Typhi* porins depended almost exclusively on IFN-gamma. The immunosuppression generated by the 4T1 cells does not affect the reactivity of the IgMs in their antigen recognition. This result had been previously reported [11]. However, in the presence of BPA, there is apparently a synergistic effect with a decrease in gamma IFN that would further decrease the concentration of IFN-gamma as demonstrated by the work of Gostner or Sawai [23,24] where the treatment of cells with BPA results in a significant and dose-dependent suppression of mitogen-induced tryptophan breakdown and neopterin formation along with a decrease in IFN-gamma levels.

In addition, the RT-PCR analysis of tumor samples showed a decreased expression of TNF-α and IFN-gamma in the group exposed to BPA [9]. On the other hand, it has also been reported that IFN-gamma inhibits the transcription of peroxisome proliferator-activated receptor gamma (PPAR-gamma) [25]. PPAR-gamma ligands can stimulate the differentiation of human B cells and promote the production of antibodies [26]. BPA is an agonist for PPAR-gamma in B cells. The activation of PPAR-gamma induces the expression of the transcription factor Blimp1, which subsequently inhibits B cell proliferation through the repression of c-myc while directing the differentiation of B terminal cells towards the IgM secreting plasma cells. Therefore, exposure to BPA results in the plasmablasts being subjected to fewer cycles of cell division before becoming plasma cells, which leads to a general decrease in the abundance of plasma cells and lower levels of circulating IgM [27].

Taking this information together, the data support a model in which BPA modifies the activation pathways of normal B-2 cells and promotes the differentiation of B-2 cells while suppressing the expansion of plasma blasts, which results in fewer plasma cell secretors of IgM. This is shown in Figure 4 where we see a decrease in IgM reactivity which is likely to be of high affinity antibodies due to the abovementioned reasons. Therefore, BPA reduces the ability of IgM recognition maybe through a decrease in the population of B-2 cells.

## 5. Conclusions

Our results demonstrate that BPA administered during the neonatal period has no impact on the percentage of the B cell population but instead affects its ability to generate high affinity IgM antibodies, as measured by its reactivity to tumor antigens.

## Figures and Tables

**Figure 1 ijerph-16-01784-f001:**
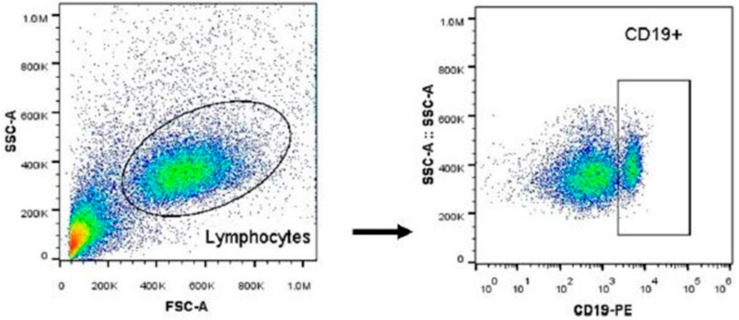
Strategy used for the identification of B lymphocytes. The dot plot is representative of the region of interest. Data analyzed with Flow Jo software.

**Figure 2 ijerph-16-01784-f002:**
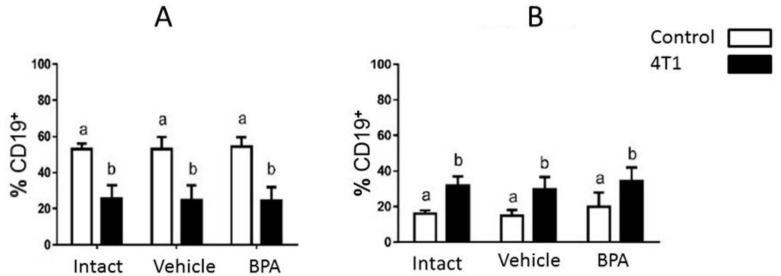
(**A**): B lymphocyte population in spleen. B lymphocytes are diminished in tumor bearing animals. (**B**): Population of B lymphocytes in draining peripheral lymph nodes. An increase in B lymphocytes is observed as a consequence of tumor development. However, for both groups, no significant differences were observed due to the neonatal treatment effect in healthy groups or in those with tumor development. Data were obtained from 3 independent experiments and are expressed as mean ± SD; n = 12 per group. Notation: (b ≠ a), *p* < 0.05.

**Figure 3 ijerph-16-01784-f003:**
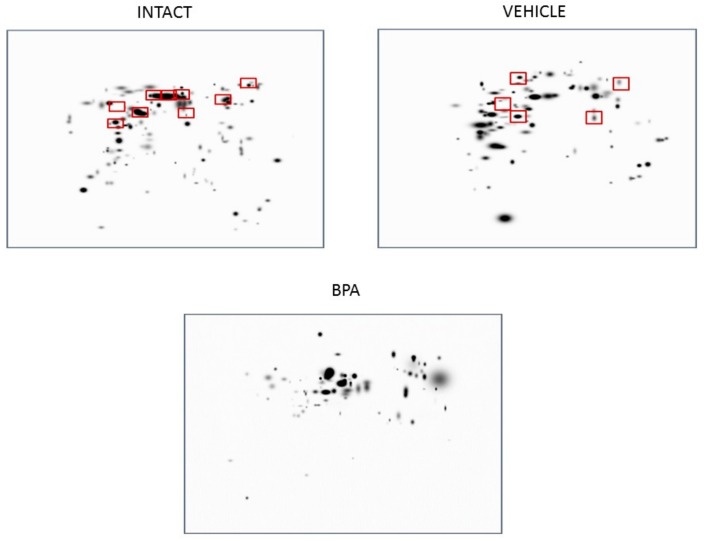
2D master images obtained from average images of mice in each experimental situation. Control: N = 6, Vehicle: N = 4 and BPA: N = 7. The red boxes indicate the spots that are common to all the mice in that group.

**Figure 4 ijerph-16-01784-f004:**
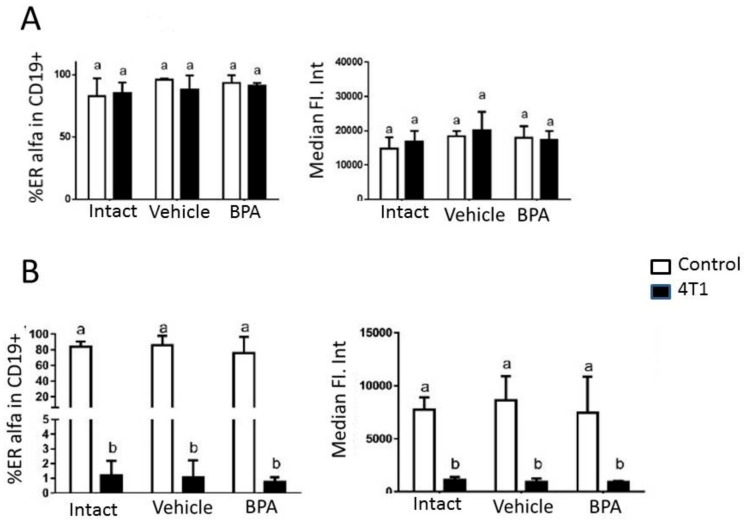
(**A**): Evaluation of ERα expression in terms of percentage of ERα^+^ cells and level of expression represented by the median intensity of fluorescence in draining peripheral lymph nodes using flow cytometry. No changes are observed between experimental groups. (**B**): Evaluation of ERα expression in terms of percentage of ERα^+^ cells and level of expression represented by the median intensity of fluorescence in spleen. It was observed that the B lymphocytes of the animals that developed tumors had a lower expression of ERα compared to healthy individuals. This phenomenon was more evident in the groups treated with vehicle and BPA. Data were obtained from 2 independent experiments and are expressed as mean ± SD; n = 10 per group. Notation: (b ≠ a), *p* < 0.05.

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
