# Peer review of "Neonatal Bisphenol A Exposure Affects the IgM Humoral Immune Response to 4T1 Breast Carcinoma Cells in Mice"

_ijerph, 2019, doi:10.3390/ijerph16101784_

Round 1

Reviewer 1 Report

Overall the manuscript and data quality are great. There are a few things I noticed:

The housing of animals need to be described in greater detail (e.g., humidity, bedding, # of mice per cage).

The selection of subcutaneous injection needs to be justified and supported by literature, as oral gavage is used more frequently for BPA administration.

It would be better to know if the mice receiving breast cancer cells actually develops breast cancer or not.

The study would be strengthened if the IgM (or other antibodies produced by B cells) were measured.

Author Response

Reviewer 1

R1.- The housing of animals need to be described in greater detail (e.g., humidity, bedding, # of mice per cage).

Regarding to the housing of animals, in the manuscript (line 68) was written that the Mexican regulation (NOM-062-ZOO-1999) was follow. Now in the text was added:    As an ethical statement, the technical specifications for the production care and use of laboratory animals can be consulted in: https://www.biomedicas.unam.mx/wp-content/pdf/unidad-de-modelos-biologicos/nom-062-zoo-1999.pdf?x21431. (Line 68)

R1.- The selection of subcutaneous injection needs to be justified and supported by literature, as oral gavage is used more frequently for BPA administration.

Regarding the tumor induction, in the text was added: “For the tumor induction the method publish by Pulaski and Ostrand-Rosenberg was follow (12)”(line 91) and further “All mice that received 4T1 cells developed a tumor.” (line 95)

Effectively as established by the reviewer, the oral gavage is used more frequently for the BPA administration. However, the text indicates “subcutaneous injection was selected instead as no difference between oral and subcutaneous routes are observed in neonate mice in this case” (line 83) For that reason we did not make any changes in the text regarding the reviewer observation.

R1.- It would be better to know if the mice receiving breast cancer cells actually develops breast cancer or not.

In the text was added now  “All mice that received 4T1 cells developed a tumor.” (line 95)

R1.- The study would be strengthened if the IgM (or other antibodies produced by B cells) were measured.

We did Immunoblots to evaluate the IgG recognition but null recognition was found. Now in the text was added “The recognition of IgG was also evaluated but a null recognition was found in the controls as in the individuals treated with the 4T1 cells.” (Line 25).  We are unfortunately unable to quantify the amount of IgM in an exact unit of measure (ie mg/ml) as one would normally attempt to do in a sandwich ELISA with a recombinant protein as a standard because the total serum amount (a few microliters) were used for immunoblot.

Reviewer 2 Report

The manuscript "Neonatal Bisphenol A Exposure Affects the IgM  Humoral Immune Response to 4T1 Breast Carcinoma  Cells in Mice" described the alterations in the humoral antitumor immune (IgM) response in adult life after a single neonatal exposure to BPA. No significant changes were  found in the B cell populations in the peripheral lymph nodes and the spleen. In all, the novelty or necessarity of the manuscript has not been well described. The experimental data have not been well discussed for in-deep discussion on the relationship between BPA exposure and  IgM  Humoral Immune Response. Additionally, language is also poor, with lots of spelling mistakes. A major revision is needed before publication.

1 The novelty in the introduction should be improved.

2 HD figure is needed.

3 The discussion should be reorganized. For example, what is the reason for both no significant changes in the B cell populations and the level of ERα expression.

4 Spelling mistakes:

Line 56, by reactivity by the reactivity

Line 204, in the in the 

5 some missing information:

Line 12, Line 259

6 No ethical statement

Author Response

Reviewer 2

R2.- The novelty in the introduction should be improved.

The reviewer raises an important point which we failed to clarify in the manuscript.  The novelty of our study establishes that there are changes in the pattern of recognition by IgM towards tumor antigens. This variation means that individuals treated with BPA stop recognizing antigens that are normally recognized by all individuals. Reference 11 cites a work that our group published in which we describe -among other things- the common antigens that are recognized by mice throughout the time that the tumor develops. We found that as the tumor develops, more antigens common to all individuals are detected. We interpret this effect as the appearance of IgM which is product of the adaptive immune response of the mouse and differentiate it from the innate immune response that occurs at time zero, that is, before the application of the tumor. We believe that BPA affects the adaptive response through the proven effects of BPA on cytokines such as INF gamma (reference 9) and the important role that IFN plays in the differentiation of B cells. This text appears now in the introduction.

R2.- HD figure is needed.

The figures were submitted as high resolution files (at least 300 dpi). We tried several formats and resolutions and there was no improvement in the data that the images provide.

R2.- The experimental data have not been well discussed for in-deep discussion on the relationship between BPA exposure and  IgM 

Evidently, with more experimental data, major depth in the discussion. With the experimental data that we have and the experimental data that our group has published (ref 9 and 11) we can put together a model that is in congruence with the bibliography. The data support a model in which BPA modifies the activation pathways of normal B-2 cells promoting the differentiation of B-2 cells while suppressing the expansion of plasmablast, resulting in fewer plasma cells secretors of IgM. We cannot go deeper because we need more experimental evidence to support each word. We are working on obtaining new data.

R2. - The discussion should be reorganized. For example, what is the reason for both no significant changes in the B cell populations and the level of Er-α expression.

A. B lymphocytes are part of the adaptive immune system that is specialized in the antibody production, which is part of the humoral immunity. It has been described that B lymphocytes have the expression of both nuclear ERs in all B cells subsets. In this sense, E2 has stimulatory effects on B differentiated lymphocytes derived from human PBMCs. It increased the immunoglobulin (Ig)G and IgM production in a dose dependent manner, this effect was enhanced by the addition of IL-10, an anti-inflammatory cytokine, to B cells previously treated with E2 the above becomes relevant in an autoimmune context. The stimulatory effect of E2 on the antibody titers has been  observed since the 80s on in vitro studies and in the serum of rats administered with this hormone, where an increase in IgM antibodies was reported. Of note, it has been reported that IgMs have a direct cytotoxic effect on transformed cells, trough the activation of the complement pathway.  This is relevant since the increases on IgMs levels due to E2 exposure are important for breast cancer suppression. Besides, they also might serve as diagnostic indicators of the phenotype or stage of this pathology, due to the fact that they are well correlated with the clinical score and disease spread of breast cancer patients, however, more studies are necessary to affirm this fact. Added to that, E2 through ERα pathway also impacts on the activation and survival of B cells through the modulation of several genes. These effects were observed in splenic B cells derived from ovariectomized mice or not, administered with it. Interestingly, these results were reverted in mice treated with ICI 182,780. Regarding to effects of GPR30 on B lymphocytes, some reports have mentioned that different chemokines can activate it triggering different roles of B subsets such as migration, chemotaxis, proliferation, and apoptosis, among others. In fact, this receptor has been correlated with different B cells malignancies such as leukemia and lymphomas. Nevertheless, more information or mechanisms of action related with this topic would be interesting in relation to the breast cancer pathogenesis.

Said that, in here, we did not  found changes in B  lymphocytes  and ERα levels, because it may be more appropriate to search for plasmatic cells. B cells did not change, because, the production of the same may not be affected by BPA, but its differentiation to plasmatic cells may be the key. We are currently working to dilucidate this effect of BPA. As for the ERα, probably the binding of BPA does not affect its expression, and  thus, there  won`t be changes on that, even the mechanism of action is through that pathway. To block ER using tamoxifen or ICI-250 would be revealing in dilucidating if BPA mechanism of action it is only by binding to ERα.  Or, it  may be that  other steroid hormone  receptors are  involved in the BPA effect, such as PR or even ERα.

 Additional text regarding this point was included now in the discussion (line 230)

R2.- Spelling mistakes and some missing information:

The spelling mistakes and the missing information were corrected.

R2.-No ethical statement

The address of Etical statement is now included in the text. (Line 68)

Round 2

Reviewer 2 Report

The authors have carefully revised the manuscript. The study would be helpful for understanding the adverse effect of BPA.  I think the manuscript can be published in the journal of International Journal of Environmental Research and Public Health .

Author Response

Thank you very much for the observations

Reviewer: For the statistics, l don’t think the disbution of the data is normal. Why do you use the parametric method? Why do you use the Shapiro–Wilk test, but your subgroup number is less than 50?

The two-way ANOVA test is used. Do you find any interactions in your variables? Please explain it.

Answer: We have reviewed our procedures and have realized that the Shapiro-Wilk test was used in a previous paper and was NOT used in the current work. We used 2-way ANOVA and Bonferroni multiple comparison between all groups and we did not find significant differences. The paragraph is now as follows:

2.6. Statistical analysis.

The experimental design considers two independent variables: neonatal exposure (Intact, BPA vehicle) and induction of mammary tumors (Control or 4T1). Data from 2–3 independent experiments were analyzed using the Prism 6® software (GraphPad Software Inc.) A one-way ANOVA (alpha = 0.05) was performed. The differences were considered to be significant if p < 0.05. The estrogen receptor expression data consider the two independent variables and therefore, a two-way ANOVA (alpha = 0.05) was performed, followed by a Bonferroni multiple comparison between all groups.

and it's marked with yellow

Reviewer: L 36: endocrine disruptering chemicals (EDCs) not endocrine disrupting compounds

Answer:  In the text now says: endocrine disruptering chemicals and it's marked with yellow

Reviewer: L 44: suggesting change as “ ....to occur by oral ingestion”

Answer: In the text now says: to occur by oral ingestion and it's marked with yellow
